# Response to Fingolimod in Multiple Sclerosis Patients Is Associated with a Differential Transcriptomic Regulation

**DOI:** 10.3390/ijms25031372

**Published:** 2024-01-23

**Authors:** Alicia Sánchez-Sanz, Rafael Muñoz-Viana, Julia Sabín-Muñoz, Irene Moreno-Torres, Beatriz Brea-Álvarez, Ofir Rodríguez-De la Fuente, Antonio García-Merino, Antonio J. Sánchez-López

**Affiliations:** 1Neuroimmunology Unit, Instituto de Investigación Sanitaria Puerta de Hierro-Segovia de Arana, 28222 Madrid, Spain; alicia.sanchez@idiphim.org; 2Bioinformatics Unit, Instituto de Investigación Sanitaria Puerta de Hierro-Segovia de Arana, 28222 Madrid, Spain; rmviana@idiphim.org; 3Department of Neurology, Hospital Universitario Puerta de Hierro Majadahonda, 28222 Madrid, Spain; julia.sabin.m@gmail.com (J.S.-M.); ofir.rodriguez@yahoo.com (O.R.-D.l.F.); 4Demyelinating Diseases Unit, Hospital Universitario Fundación Jiménez Díaz, 28040 Madrid, Spain; irenemt20@gmail.com; 5Radiodiagnostic Division, Hospital Universitario Puerta de Hierro Majadahonda, 28222 Madrid, Spain; beatrizbreaalvarez@yahoo.es; 6Department of Medicine, Universidad Autónoma de Madrid, 28049 Madrid, Spain; 7Red Española de Esclerosis Múltiple (REEM), 08028 Barcelona, Spain; 8Biobank, Instituto de Investigación Sanitaria Puerta de Hierro-Segovia de Arana, 28222 Madrid, Spain

**Keywords:** multiple sclerosis, fingolimod, transcriptome, biomarker, disease-modifying therapies, treatment response, RNA sequencing

## Abstract

Fingolimod is an immunomodulatory sphingosine-1-phosphate (S1P) analogue approved for the treatment of relapsing-remitting multiple sclerosis (RRMS). The identification of biomarkers of clinical responses to fingolimod is a major necessity in MS to identify optimal responders and avoid the risk of disease progression in non-responders. With this aim, we used RNA sequencing to study the transcriptomic changes induced by fingolimod in peripheral blood mononuclear cells of MS-treated patients and their association with clinical response. Samples were obtained from 10 RRMS patients (five responders and five non-responders) at baseline and at 12 months of fingolimod therapy. Fingolimod exerted a vast impact at the transcriptional level, identifying 7155 differentially expressed genes (DEGs) compared to baseline that affected the regulation of numerous signaling pathways. These DEGs were predominantly immune related, including genes associated with S1P metabolism, cytokines, lymphocyte trafficking, master transcription factors of lymphocyte functions and the NF-kB pathway. Responder and non-responder patients exhibited a differential transcriptomic regulation during treatment, with responders presenting a higher number of DEGs (6405) compared to non-responders (2653). The S1P, NF-kB and TCR signaling pathways were differentially modulated in responder and non-responder patients. These transcriptomic differences offer the potential of being exploited as biomarkers of a clinical response to fingolimod.

## 1. Introduction

Multiple sclerosis (MS) is a chronic demyelinating disease of the central nervous system (CNS) characterized by the autoimmune attack of the myelin sheath, which leads to axonal damage and neurodegeneration. The early stages of the disease are characterized by the infiltration of immune cells from the periphery into the CNS through a leaking blood–brain barrier [1]. These immune cells comprise autoreactive T and B lymphocytes and innate immune cells such as macrophages and natural killer (NK) cells. T lymphocytes are considered the main effectors in the pathogenesis of MS, including CD4+ and CD8+ T lymphocytes and, in particular within the CD4+ subset, Th1 and Th17 effector T cells. B lymphocytes produce antibodies that can be detected in the cerebrospinal fluid of MS patients and contribute to the pathogenesis of the disease through antigen presentation and cytokine production [2].

Current therapies in MS are based on immunomodulatory or immunosuppressant drugs known as disease-modifying therapies (DMTs), which aim to prevent the infiltration of immune cells from the periphery into the CNS parenchyma [3]. An expanding number of DMTs have been approved in recent years, with different mechanisms of action, routes of administration and safety and efficacy levels. This has led to an increasing complexity in the clinical practice in selecting the appropriate DMT for each patient. Patients can present a suboptimal response to a DMT, which results in the appearance of relapses and magnetic resonance imaging (MRI) lesions that lead to the accumulation of disability [4]. Therefore, the identification of biomarkers of clinical response is a major necessity in MS to classify patients as optimal responders to a DMT and avoid the use of ineffective drugs in non-responders.

Fingolimod (Gilenya^®^, Novartis, Basel, Switzerland; also known as FTY720) is a sphingosine 1-phosphate (S1P) analogue approved for the relapsing-remitting form of MS (RRMS). Fingolimod acts as a non-selective functional antagonist on the S1P receptors (S1PR) S1PR1 and S1PR3-5, S1PR1 being the most important isoform for the achievement of immunomodulatory effects in the periphery. S1PR1 is expressed on the surface of lymphocytes and regulates their egress from secondary lymphoid organs (SLOs) into the bloodstream through the detection of S1P concentration gradients. The binding of fingolimod to S1PR1 causes its internalization, leaving lymphocytes unresponsive to S1P egress signals, which results in lymphocyte retention inside SLOs, preventing the entrance of inflammatory cells into the CNS [5].

Fingolimod was the first S1PR modulator approved for MS therapy, and it is widely prescribed worldwide due to its advantages over other DMTs. Fingolimod is an oral drug, which fosters therapeutic adherence, and it presents a moderate efficacy in the control of the disease with a manageable safety profile [6,7]. Beyond lymphocyte retention in SLOs, which results in an overall decrease in absolute lymphocyte counts, fingolimod exerts specific changes in the composition of T and B cell subsets in peripheral blood mononuclear cells (PBMCs) of MS-treated patients. On T lymphocytes, fingolimod reduces primarily the percentages of cells expressing CCR7, a homing receptor that promotes lymphocyte retention inside SLOs, diminishing naïve and central memory T cells (both CCR7+), and leaving effector memory T cells (CCR7-) unaltered [8,9,10]. On the B cell compartment, fingolimod reduces the percentages of memory B cells and increases naïve and transitional B cells and plasmablasts [11,12,13,14,15].

The study of PBMC subpopulations has been commonly used in MS research with the aim of identifying biomarkers of clinical response to DMTs, including fingolimod. Although some associations between specific subpopulations of PBMCs and clinical response to fingolimod have been found [16,17,18,19,20,21], results between studies are heterogeneous, hindering the validation of biomarkers with applicability in the clinical practice and highlighting the necessity of searching for novel biomarkers beyond immune cell subsets. A previous study from our group combined the study of cellular subpopulations with transcriptome profiling and identified a baseline signature with potential to predict clinical response to fingolimod [22]. In addition, a differential cellular and transcriptomic regulation was observed between responder and non-responder patients at 6 months of fingolimod therapy [22]. The aim of the present study was to deepen our knowledge on the distinctive regulation between responder and non-responder patients during fingolimod treatment by studying the effect of this DMT at the transcriptional level at 12 months of therapy and its association with clinical response. This could shed light on the molecular mechanism of action of fingolimod’s therapeutic failure and guide the identification of specific biomarkers of clinical response.

## 2. Results

### 2.1. Clinical Response to Fingolimod

The demographic and clinical characteristics at baseline of our cohort of MS patients and healthy donors (HD) are summarized in Table 1. The mean age of MS patients was 37 years and 60% were female, characteristics like those of the group of HD. MS patients had a mean disease duration of 6 years and a mean score of around one on the expanded disability status scale (EDSS). All the patients had received at least one previous DMT, switching most of them from natalizumab (*n* = 6) due to John Cunningham virus antibody positivity, while the rest had switched from interferon beta (IFN-β) (*n* = 2) or dimethyl fumarate (DMF) (*n* = 2) due to a lack of efficacy. Appendix A and Appendix A show the detailed characteristics of each subject in the group of HD and MS patients, respectively.

At 2 years of fingolimod therapy, five patients achieved no evidence of disease activity-3 (NEDA-3) and were classified as responders, while the remaining five patients presented evidence of disease activity-3 (EDA-3) and were classified as non-responders (Table 2). Within the non-responders, one patient had experienced confirmed disease progression (CDP), two patients had experienced relapses, one patient presented MRI activity and relapses and one patient had experienced relapses, CDP and MRI activity. Responder and non-responder patients to fingolimod presented similar baseline characteristics (Table 1).

### 2.2. Transcriptomic Profile of HD and MS Patients

Principal component analysis (PCA) of the transcriptomic profile of HD and MS patients before and after fingolimod therapy revealed three differentiated clusters of samples. MS patients and HD presented a clearly differentiated transcriptomic profile, grouping separately in two different regions of space (Figure 1A). This separation between MS patients and HD was clear for all the samples and was independent of the time of fingolimod treatment, as both the baseline and 12-month samples separated distinctly from HD. In the HD group, all the subjects clustered together, while in the group of MS patients, there was one baseline sample that positioned in a notably distanced region of space compared to the rest of MS samples, and which was excluded from subsequent analyses. However, the sample from this patient at 12 months of therapy with fingolimod clustered with the rest of MS samples at 12 months and was not excluded from the study.

In addition, the baseline samples from MS patients formed a differentiated cluster from the 12-month samples, indicating that fingolimod treatment significantly affects the gene expression profile (Figure 1A). This differential clustering was clear except for one 12-month sample which grouped with the baseline samples, and which corresponded to a non-responder patient. Within MS patients, responder and non-responder patients did not form differentiated clusters at baseline, indicating that there is not a discernible transcriptomic profile between these subgroups before treatment. However, at 12 months of therapy, although they did not form separated clusters either, samples from responder patients grouped together while samples from non-responders distributed in a more scattered way, suggesting a more homogeneous transcriptomic response to fingolimod in responder patients.

Regarding the number of differentially expressed genes (DEGs) between MS patients and HD, we identified 6474 DEGs (3440 upregulated and 3034 downregulated) in MS patients at baseline compared to HD (Figure 1B and Appendix A). At 12 months of fingolimod therapy, the number of DEGs compared to HD increased to 10,170 (5137 upregulated and 5033 downregulated), indicating that fingolimod transcriptionally differentiates MS patients from HD even more (Figure 1B and Appendix A). From these DEGs, 4825 were common at baseline and 12 months of therapy, suggesting that fingolimod modifies mainly additional genes, leaving most of the dysregulated genes at baseline compared to HD unaltered (Figure 1B). Considering the clinical response to fingolimod, responder and non-responder patients presented a similar number of DEGs (5116 and 5255 DEGs, respectively) at baseline compared to HD (Figure 1C and Appendix A). From these DEGs, 3788 were common between responders and non-responders at baseline (Figure 1C). However, at 12 months of fingolimod therapy, the number of DEGs compared to HD was higher in responders than in non-responders (9408 and 8361 DEGs, respectively, of which 6983 were common between responders and non-responders) (Figure 1C and Appendix A).

### 2.3. Effect of Fingolimod Treatment on Gene Expression

A total of 7155 genes were differentially expressed at 12 months of fingolimod treatment compared to baseline (Figure 2A and Appendix A). From these DEGs, 3509 were upregulated and 3646 were downregulated. Figure 2B shows the top 100 differentially expressed genes by fingolimod treatment. As foreseen, downregulation of the *S1PR1* gene, the established molecular target of fingolimod, was observed at 12 months of therapy [23]. Modulation of the remaining S1PR-coding genes was also detected; *S1PR2*, *S1PR3* and *S1PR5* were upregulated, while *S1PR4* was found to be downregulated. Additional genes implicated in S1P metabolism were found to be upregulated, such as *SPTLC1*, *DEGS1*, and *SPHK1* and *SGPL1*, which encode the enzymes responsible of synthetizing and degrading S1P, respectively.

In line with the mechanism of action of fingolimod in the alteration of lymphocyte trafficking, we observed a downregulation of SLOs homing receptor genes such as *CCR7*, *SELL* (which encodes L-selectin), *CXCR5* and *CCR6* [24]. We also detected the downregulation of markers of T and B lymphocytes including *CD3D*, *CD3E*, *CD3G*, *CD8B*, *CD19*, *CD27* and *CD38*, and the upregulation of NK cell markers such as *KLRC1*, *KLRC3*, *GZMB* and *PRF1*, in accordance with the observed effects of fingolimod on PBMC subpopulations [22]. Notably, we detected the modulation of a vast number of genes coding for cytokines and their receptors, particularly interleukins, interferons and transforming growth factors. Among the downregulated cytokine-related genes were *IL2*, *IL16*, *IL17B*, *IL7R*, *IL21*, *IL22*, *IL23A*, *TNFRSF1B*, *TNFRSF8*, *TNFRSF10D*, *TNFSF15*, *TNFRSF21* and *TRAF5*. The upregulated cytokines included *IL1RN*, *IL10*, *IL10RA*, *IL10RB*, *IL18*, *IL27*, *IFNGR2*, *IFNAR2*, *IFNGR1*, *IFNAR1*, *IFNG*, *TNFSF10*, *TRAF6*, *TNFAIP1*, *TNFRSF14* and *TNFAIP6*.

Interestingly, several transcription factors considered as master regulators of the differentiation and functions of distinct lymphocyte subsets were modulated by fingolimod treatment. Among these, we found an upregulation of the transcription factors *EOMES*, *STAT3*, *TBX21*, *SPI1*, *RUNX1*, *BCL6*, *AHR* and *NFAT5*, and a downregulation of *FOXP3*, *IKZF1*, *NFTAC1*, *FOXO1*, *EBF1*, *PAX5* and *LEF1*. In addition, we also observed that fingolimod induces a general inhibitory effect on the pathway of the inflammatory master regulator NF-kB, upregulating inhibitory proteins such as *IKBKG*, *NKIRAS1*, *NKIRAS2*, *NFKBIB*, *NFKB1*, *TRAF4* and *PRKCZ*. The expression of the Toll-like receptors *TLR1*, *TLR2*, *TLR3*, *TLR4*, *TLR8* and *TLR10* was also modulated by fingolimod. The transcription factor Nrf2 (encoded by the *NFE2L2* gene) and its downstream targets *HMOX1*, *SOD2*, *FTL* and *FTH1*, implicated in antioxidant responses, were found upregulated by fingolimod treatment.

Gene set enrichment analysis (GSEA) revealed enrichment of several gene sets by fingolimod treatment, which were mainly related to immunity (Figure 2C). Among them, we could observe enrichment of gene sets related to the complement cascade, Fc antibody receptors signaling, signaling by the BCR and by the TCR, phagocytosis, signaling by TGF-β and the CTLA4 immune checkpoint pathway. Fingolimod treatment also produced the enrichment of signaling pathways including the Wnt/β-catenin pathway, the Slit/Robo pathway, the MAPK/ERK pathway and several pathways related to apoptosis including p53 and BAD. Pathways related to the metabolism of the intracellular messengers DAG, IP3 and Ca2+ were also enriched by fingolimod treatment.

### 2.4. Gene Expression in Responder and Non-Responder Patients to Fingolimod

At baseline, only 10 genes were differentially expressed between responder and non-responder patients to fingolimod, in agreement with the PCA analysis, where there was no transcriptomic clustering of the patient samples according to their clinical response (Figure 3A and Appendix A). From these DEGs, 10 were downregulated and 1 was upregulated in responder patients compared to non-responders. We found associations between 3 of the 11 DEGS (*RMRP*, *EDIL3* and *GSTM1*) and MS in the literature [25,26,27]. The differences in these DEGs at baseline between responders and non-responders were not maintained at 12 months of therapy. At 12 months of fingolimod treatment, responder and non-responder patients presented 40 different DEGs (Figure 3 and Appendix A). From these DEGs, 30 were downregulated and 10 upregulated in responder patients compared to non-responders. Interestingly, the IL7 receptor gene (*IL7R*), which is associated with a higher risk of MS, was downregulated in responder patients [28]. One DEG implicated in S1P metabolism (*PLPP2*, enzyme that dephosphorylates S1P) also presented a lower expression in responder patients compared to non-responders at 12 months of fingolimod therapy.

When we analyzed the changes in gene expression between 12 months of fingolimod therapy and baseline in each group, we found a differential modulation during treatment between responder and non-responder patients. Responder patients presented a higher number of DEGs (6405 DEGs from which 3111 were upregulated and 3294 were downregulated) compared to non-responders, which only presented 2653 DEGs (1204 upregulated and 1451 downregulated) (Figure 3B,C and Appendix A). Interestingly, the downregulation of *S1PR1* was only observed in the responder group, which also presented a higher number of DEGs related to the S1P pathway compared to non-responders. Responders also presented a higher number of DEGs related to cytokines and to master regulator transcription factors compared to non-responders, in which the modulation of *EOMES*, *TBX21*, *SPI1*, *BCL6*, *AHR* and *EBF1* was absent. In addition, we observed the regulation of genes related to TCR signaling such as *CD3G*, *CD3D*, *CTLA4*, *LAT*, *LCK*, *ITK*, *JAK3* and *ZAP70* only in the responder group. It is also worth highlighting that the modulation of the NF-kB pathway, with genes such as *RELB*, *NFRKB*, *NFKBIB* and *IKBKG*, was also present only in the responder group.

## 3. Discussion

In this study, we describe the transcriptomic changes induced by fingolimod in PBMCs from RRMS patients at 12 months of therapy, and their association with clinical response at 2 years. To the best of our knowledge, this is the first study evaluating the long-term effect of fingolimod therapy at the transcriptional level. The transcriptomic profile of MS patients at 12 months of fingolimod therapy was found considerably modified, identifying 7155 DEGs compared to baseline. A similar number of DEGs (7546) was also found in PBMCs at 6 months of therapy [22], indicating that the gene expression profile remains stable between 6 and 12 months of fingolimod treatment. Additional transcriptomic studies at 6 months of fingolimod therapy identified a lower number of DEGs compared to baseline [29,30]. However, these studies were performed in specific subsets of PBMCs such as CD3+ T lymphocytes (555 DEGs) and monocytes (60 DEGs) and applied a fold change threshold to filter genes [29,30]. In our study, we considered DEGs only based on statistical significance without considering the magnitude of the fold change, allowing us to identify a higher number of DEGs. In addition, with our experimental design we cannot discern the contribution of each PBMC subpopulation to the observed transcriptomic changes and can only account for a global effect. In this regard, a study performed at 3 months of fingolimod treatment identified CD4+ T lymphocytes as the main subpopulation transcriptionally modified (6489 DEGs), followed by CD8+ T lymphocytes (861 DEGs) and B lymphocytes (42 DEGs), while NK cells and monocytes were unaffected [31]. The similar number of DEGs found in CD4+ T lymphocytes compared to our study suggests that this subpopulation could be the main contributor to the observed transcriptomic changes in PBMCs at 12 months. However, this remains to be elucidated as fingolimod not only affects the trafficking of T lymphocytes but it also influences B lymphocytes and NK cells [22].

Fingolimod modified a vast number of genes related to a variety of signaling pathways, thus affecting numerous cellular processes, especially immune related. We could verify the modification of genes from the S1P pathway, according to the established mechanism of action of fingolimod, including the downregulation of *S1PR1* [32]. In addition, several homing receptor genes, such as *CCR7*, were also downregulated in peripheral blood due to the retention, in absence of the competing egress-promoting signals of S1P, of cells expressing retention markers inside SLOs [33]. In line with the effect of fingolimod on the blood cellular composition, we also observed the downregulation of markers of T and B lymphocytes, and an upregulation of NK cell markers [22].

Interestingly, we have observed that fingolimod exerts important immunomodulatory effects beyond the retention of lymphocytes inside SLOs, by modifying key inflammatory mediators. These included the signaling cascades downstream of important immune receptors such as the TCR, the BCR and TLRs, the regulation of transcription factors that act as master regulators of lymphocytes and the regulation of the NF-kB and Wnt/β- catenin pathways. Activation of the Wnt/β-catenin pathway has been directly linked to suppression of inflammation in the mouse model of MS, experimental autoimmune encephalomyelitis, by promoting a regulatory phenotype in dendritic cells that leads to diminished Th1 and Th17 responses, which could be associated with the decreased percentages of Th1 and Th17 cells observed in MS patients treated with fingolimod [22,34]. Additional intracellular signaling pathways modulated by fingolimod included the MAPK/ERK pathway, the Slit/Robo pathway and the p53 pathway. The modulation of these pathways was translated in the modification of processes such as apoptosis, cytokine production, phagocytosis and lymphocyte chemotaxis. Previous studies at 3 and 6 months of fingolimod therapy also observed the modulation of similar inflammatory pathways, including cytokine production, T cell activation and chemotaxis, indicating that the modulation of specific pathways by fingolimod is stable during at least the first 12 months of therapy [22,29,35].

In addition, we analyzed the transcriptomic profile of responder and non-responder patients to fingolimod with the aim of identifying differences that could be used as predictive biomarkers of clinical response. At baseline, differences between responder and non-responder patients were very modest, with only 11 DEGs between these subgroups. From these DEGs, we found studies in the context of MS for only three of them (*RMRP*, *EDIL3* and *GSTM1*), but, interestingly, the existing studies describe a differential expression of these genes in MS patients compared to HDs, suggesting their potential use as disease biomarkers [25,26,27]. To the best of our knowledge, there are no studies evaluating the expression of these genes in relation to DMTs, but our results show their potential use as biomarkers of clinical response to fingolimod, which should be further investigated.

The number of studies correlating gene expression levels with clinical response to fingolimod is very low. In this regard, a previous study in CD8+ lymphocytes found no baseline transcriptional differences between patients with or without relapses during fingolimod treatment [36], while another study on monocytes identified only 4 DEGs at baseline between NEDA-3 and EDA-3 patients [30], none of which coincides with the 11 DEGs identified in this study. Furthermore, in a previous study from our group, we identified 36 DEGs before fingolimod treatment between NEDA-3 and EDA-3 patients [22], none of which coincides with those identified in this study. This discrepancy with our previous results could be due to the different characteristics of the cohorts of MS patients, which had a shorter disease duration, a shorter time of therapy with previous DMTs and a lower EDSS in the present study. However, the existing studies, together with the results presented in this article, seem to coincide with the fact that responder and non-responder patients to fingolimod present minimal differences at baseline, as shown by the low number of DEGs identified in all the studies [22,30,36]. In addition, the characteristics of the cohort of patients could be a decisive factor in the validation of DEGs across studies, especially if we consider that all the transcriptomic analyses correlated with clinical response to fingolimod have been performed in small cohorts of patients that could present selection bias and not be representative of the study population.

Despite the few differences at baseline, we were able to identify a clear differential transcriptomic response during fingolimod treatment according to clinical response, with responders presenting a higher number of DEGs (6405) compared to non-responders (2653). Similar results were found in our previous study, in which we identified a higher number of DEGs in responders compared to non-responders during the first 6 months of fingolimod therapy (approximately 2500 DEGs compared to 1500 DEGs, respectively) [22]. In addition, differential transcriptomic responses to the DMT DMF have also been described in PBMCs in two independent studies, also identifying a higher number of DEGs in responders compared to non-responders [37,38]. These results suggest that the changes in gene expression during treatment can reflect clinical response to DMTs; thus, they have the potential of being used as biomarkers. Although these changes would not have the advantage of predicting clinical response before starting the DMT, they could be studied during the first year of therapy to predict the clinical response in the long term. During DMF treatment, differences between responders and non-responders could already be observed at 6 weeks of therapy [38]. With fingolimod, additional studies are required to establish the minimum time needed to observe the transcriptomic differences between responder and non-responder patients so that the shortest time of treatment can be used to predict clinical response. In this regard, the transcriptomic differences between responders and non-responders could already be observed at 6 months of fingolimod therapy, although these were more pronounced at 12 months, as reflected by the greater differences in the number of DEGs at 12 months [22]. However, there are no studies evaluating the changes in gene expression in responders and non-responders to fingolimod in a period shorter than 6 months.

Deepening our knowledge of the differences observed during fingolimod therapy between responders and non-responders could shed light on the molecular mechanisms of the therapeutic failure of fingolimod and guide the search of biomarkers more precisely. One of the observed differences between these subgroups of patients was the modulation of the S1P pathway, which was more pronounced in responder patients as observed by the higher number of DEGs related to this pathway compared to non-responders. This could be indicating that non-responder patients are intrinsically less responsive to fingolimod due to a worse capacity to modulate the S1P pathway. This is suggested by the fact that the downregulation of *S1PR1* was not observed in non-responders. In addition, the enzyme *SPHK1*, which phosphorylates fingolimod to its active form, was found upregulated in both subgroups, but this upregulation was greater in responders than in non-responders [23]. This differential modulation of genes from the S1P pathway between responders and non-responders at 12 months of therapy is a novel finding that could be relevant for monitoring the efficacy of fingolimod due to its direct relationship with the established mechanism of action of the drug [23]. Moreover, these differences were not observed at 6 months of fingolimod therapy, indicating that loss of modulation of the S1P pathway could be occurring in non-responder patients after the first 6 months of therapy [22].

Additionally, we have observed that the modulation of the NF-kB pathway was only present in responder patients. This differential modulation was already described at 6 months of fingolimod therapy and, interestingly, transcriptional modulation associated with the inhibition of the NF-kB pathway was also found in responder patients to DMF but not in non-responders [22,37,38]. NF-kB is a master regulator of inflammation that is directly linked with the pathophysiology of MS; thus, the modulation of the NF-kB pathway is of relevance in the therapeutic of this disease [39]. The possibility exists that the ability to modulate this pathway could be responsible for clinical response to fingolimod in responder patients, and it should be further studied as a candidate biomarker in a focused search. Moreover, the modulation of the TCR signaling pathway that was observed only in the responder group represents a novel finding that is directly linked to the regulation of NF-kB through the triggering cascade that leads to its nuclear translocation and represents another potential biomarker of response to fingolimod [40].

The main limitation of this study is the small sample size of our cohort of patients, especially of the subgroups of patients according to their clinical response to fingolimod. In addition, our data are only descriptive, and the identified DEGs in this study should be confirmed in an independent cohort of patients. Our findings provide insight into the mechanism of action of fingolimod, which is of relevance for the monitoring of its therapeutic efficacy and safety in patients, as well as for the development of new therapies in MS. In addition, we have identified specific pathways that are differentially regulated between responder and non-responder patients to fingolimod, and which offer the potential of being exploited as biomarkers of clinical response with the ultimate goal of providing a personalized medicine.

## 4. Materials and Methods

### 4.1. Patients and Healthy Donors

Enrollment to the study was limited to patients with a diagnosis of RRMS according to McDonald criteria and with an indication for treatment with fingolimod [41,42]. Patients who had received steroid treatment in the previous month or immunosuppressants in the previous year were excluded. Ten patients from the Multiple Sclerosis Unit at Hospital Universitario Puerta de Hierro Majadahonda were included in the study. Patients switching from natalizumab had a 1–2 months washout period before starting therapy with fingolimod. Patients switching from IFN-β or DMF did not need a washout period according to the approved guidelines. In compliance with the approved indications of use for fingolimod, patients were treated with 0.5 mg daily [42]. Lithium heparin tubes (Greiner Bio-One, Frickenhausen, Germany) were used to collect venous blood samples from each patient immediately before starting fingolimod and at 12 months of therapy. PBMCs were isolated from peripheral blood samples by Ficoll-Paque^®^ density gradient centrifugation and cryopreserved in liquid nitrogen until use. A cohort of 10 HDs with samples at a single time point, which had been published in a previous study from our group, was also included in this study [37].

### 4.2. Clinical and MRI Measures

Clinical outcomes were evaluated in all patients at baseline and at 1 and 2 years of fingolimod therapy. Data from EDSS scores and the number of clinical relapses were collected. The increase of 0.5 or more EDSS points maintained for 3 months was considered as CDP. A 1.5T brain MRI was also performed at 1 and 2 years of fingolimod therapy to obtain the number of gadolinium-enhanced T1 lesions (Gd+) and the number of new or enlarged T2-weighted lesions (T2w). MRI activity was defined as the appearance of new Gd+ and/or T2w lesions. NEDA-3 status was defined as the absence of MRI activity, clinical relapses, and CDP, according to the published literature [19]. As most of the patients were switching from natalizumab, which has been associated with a risk of MS reactivation shortly after fingolimod initiation, we decided to assess clinical response at a long time period (2 years) to obtain more robust results regarding the clinical response [43]. Patients who achieved NEDA-3 at 2 years of fingolimod therapy were classified as responders. Patients who did not achieved NEDA-3 at 2 years (EDA-3) were classified as non-responders to fingolimod.

### 4.3. RNA Isolation and Sequencing

The Maxwell^®^ 16 LEV simply RNA Cells kit and the robotic platform Maxwell^®^ 16 (both from Promega Biotech Ibérica, Madrid, Spain) were used for the extraction of total RNA from PBMCs, following the instructions by the manufacturer. The concentration of the extracted RNA was assessed using the Qubit^®^ 3.0 fluorometer (Thermo Fisher Scientific, Waltham, MA, USA). The quality of the RNA was measured using the 2100 Bioanalyzer system (Agilent Technologies, Santa Clara, CA, USA), and all the samples presented an RNA integrity number above 9. From the total RNA, we isolated the mRNA fraction based on the poly (A) tail by binding poly (T) oligos attached to magnetic beads. The isolated mRNA was chemically fragmented, and reverse transcribed to cDNA. An end-repair process was applied to the cDNA fragments, which were also added a single ‘A’ base and ligated to the adapters. The indexed library of double-stranded DNA was obtained after purification and enrichment by PCR. The 4200 TapeStation system (Agilent Technologies) was used to examine the quality and quantity of the libraries. Sequencing was performed in the NovaSeq 6000 Sequencing System (Illumina, San Diego, CA, USA) by pair-end (150 × 2) and at a sequencing depth of 40 million reads per sample.

### 4.4. Bioinformatics Analysis

Quality control of the raw reads was conducted with fastqc v0.12.1. Adapter sequences were removed with cutadapt v4.4 [44]. Trimmed reads were further selected with trimmomatic v0.39 for size bigger than 20nt and mapq > 30 [45]. Only reads that satisfied these criteria were used in ulterior analyses. Reads were aligned with HISAT2 v2.2.1 to the human genome (GRch38 primary assembly). Mapped reads were annotated to genes from Ensemble (GRch38 v109) and quantified using HTseq v2.0.3 [46]. Gene counts were analyzed using principal components with custom R scripts (R version 4.3.1). The sample responder at baseline number 5 showed abnormal clustering and was rejected for further analyses. Differential expression analysis was performed using DESeq2 v1.40.2 [47]. GSEA was performed using the standalone application GSEA v4.3.2 [48].

## 5. Conclusions

Our study describes the transcriptomic changes induced by fingolimod in PBMCs at 12 months of therapy and their association with clinical response. Fingolimod modified a vast number of genes beyond the S1P pathway, which were mainly immune-related, including cytokines, the NF-kB pathway and master transcription factors of inflammation. In addition, responder and non-responder patients to fingolimod exhibited a differential transcriptomic regulation during treatment, with responders presenting a higher number of DEGs compared to non-responders. Relevant pathways for the therapeutic efficacy of fingolimod, including the S1P, NF-kB and TCR signaling pathways were differentially modulated in responder patients compared to non-responders, offering the potential of being exploited as biomarkers of clinical response to fingolimod.

## Figures and Tables

**Figure 1 ijms-25-01372-f001:**
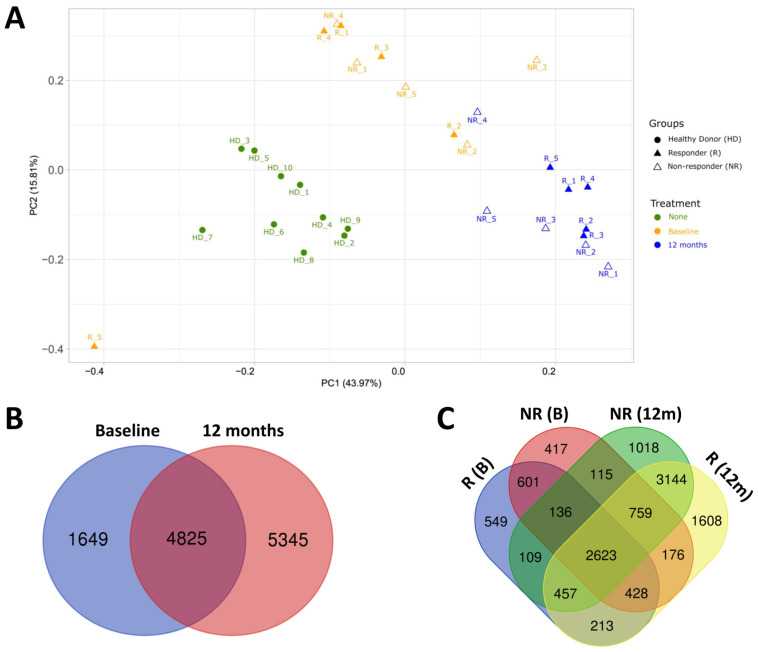
Transcriptomic profile of healthy donors (HD) and multiple sclerosis (MS) patients. (**A**) Principal component analysis of the gene expression profile of HD and MS patients at baseline and at 12 months (12m) of fingolimod therapy. (**B**) Venn diagram indicating the number of differentially expressed genes (DEGs) in MS patients at baseline and at 12m of fingolimod therapy compared to HD. (**C**) Venn diagram showing the number of DEGs in responder (R) and non-responder (NR) patients to fingolimod at baseline and at 12m of therapy compared to HD.

**Figure 2 ijms-25-01372-f002:**
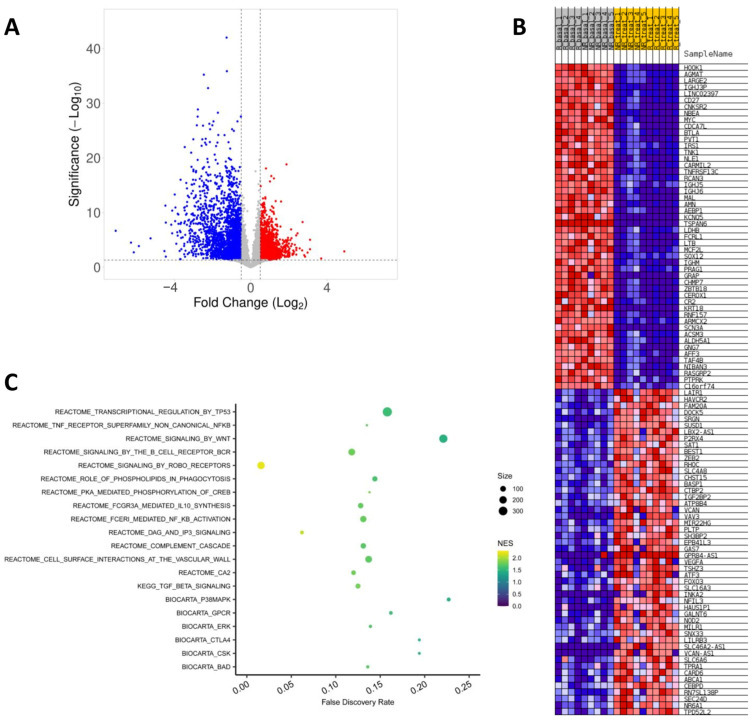
Effect of fingolimod treatment on gene expression. (**A**) Volcano plot of the differentially expressed genes (DEGs) at 12 months of fingolimod therapy compared to baseline. DEGs with adjusted *p*-value < 0.05 and fold changes >0.5 or <−0.5 are represented. Upregulated, downregulated and genes with no significant changes or smaller fold changes are represented by red, blue and grey dots, respectively. (**B**) Top 100 DEGs by fingolimod therapy. The heatmap illustrates the 50 most highly upregulated genes (red) and downregulated genes (blue) in multiple sclerosis patients at baseline (**left**) and at 12 months of fingolimod therapy (**right**). (**C**) Gene set enrichment analysis of DEGs by fingolimod treatment. The x-axis represents the false discovery rate of each gene set. The color and size of the dots represent the normalized enrichment score (NES) and the number of DEGs mapped to the indicated pathways, respectively.

**Figure 3 ijms-25-01372-f003:**
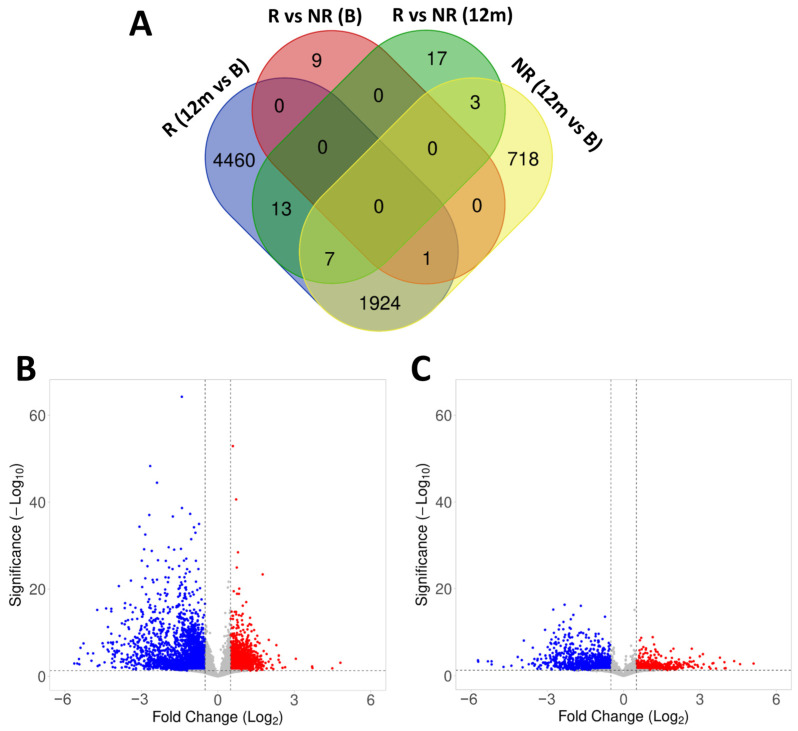
Gene expression in responder (R) and non-responder (NR) patients to fingolimod. (**A**) Venn diagram indicating the number of differentially expressed genes (DEGs) between R and NR to fingolimod at baseline and at 12 months of therapy. (**B**) Volcano plot of the DEGs at 12 months of fingolimod therapy compared to baseline in R patients. (**C**) Volcano plot of the DEGs at 12 months of fingolimod therapy compared to baseline in NR patients. (**B**,**C**) DEGs with adjusted *p*-value < 0.05 and fold changes >0.5 or <−0.5 are represented. Upregulated, downregulated and genes with no significant changes or smaller fold changes are represented by red, blue and grey dots, respectively.

**Table 1 ijms-25-01372-t001:** Baseline demographic and clinical characteristics of multiple sclerosis patients and healthy donors.

	Groups		MS Subgroups	
	HD (*n* = 10)	MS (*n* = 10)	Responders (*n* = 5)	Non-Responders (*n* = 5)
**Age (years) ^1^**	43.30 ± 10.60	36.70 ± 11.03	40.00 ± 13.32	33.40 ± 8.29
**Sex (% of female) ^2^**	70%	60%	40%	80%
**Disease duration (years) ^1,3^**		6.35 ± 4.94	6.32 ± 5.02	6.38 ± 5.45
**Time since DMT onset (years) ^1^**		3.38 ± 3.73	2.55 ± 1.84	4.20 ± 5.11
**Nº of previous DMTs ^1^**		1.40 ± 0.70	1.20 ± 0.45	1.60 ± 0.89
**Immediately previous DMT**				
** ** **Natalizumab**	60%		60%	60%
** ** **Interferon beta**	20%		20%	20%
** ** **Dimethyl fumarate**	20%		20%	20%
**EDSS ^1^**		1.60 ± 1.84	2.10 ± 2.38	1.10 ± 1.14

^1^ Values are the mean ± standard deviation of each group. The Mann–Whitney test was used to compare differences between healthy donors (HD) and multiple sclerosis patients (MS), and, within the MS group, between responder and non-responder patients to fingolimod. ^2^ Chi-Square test was used to compare two proportions. ^3^ Time since the first symptoms of MS. *p* < 0.05 was considered statistically significant. DMT, disease-modifying treatment; EDSS, expanded disability status scale.

**Table 2 ijms-25-01372-t002:** Clinical outcomes in non-responder patients to fingolimod.

EDA-3 Patients	MRI Activity	Nº of Relapses	CDP
NonResponder_1	No	1	No
NonResponder_2	No	0	Yes
NonResponder_3	Yes	1	No
NonResponder_4	Yes	1	Yes
NonResponder_5	No	1	No

MRI, magnetic resonance imaging; CDP; confirmed disease progression; EDA-3, evidence of disease activity 3.

## Data Availability

Data are presented in the manuscript and in the Appendix A. The complete raw RNA-sequencing data of HD and of MS patients are deposited in the NCBI’s public repository Gene Expression Omnibus (GEO) and are accessible through the accession numbers GSE235357 and GSE250453, respectively.

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
