# Peer review of "Response to Fingolimod in Multiple Sclerosis Patients Is Associated with a Differential Transcriptomic Regulation"

_ijms, 2024, doi:10.3390/ijms25031372_

Round 1

Reviewer 1 Report

Comments and Suggestions for Authors

The manuscript presented here is really thorough, the study is adequately designed, and the results provided in the manuscript are clear and convincing. I only have minor comments

The introduction provide enough background, but I would suggest to add few lines about the role of the immune system in the MS since most of the gene-enrichment data are about immune system. 

Also, In the results or discussion I would suggest to provide more insights about WNT-B Catenin pathway. We know this pathway is crucial in several diseases. What are the consequences of the regulation of this pathway by the Fingolimod. 

Please in the discussion, state the limitation of this study and discuss what your results can bring to the field and what should we improve to provide better solutions for patients. 

in the materials and methods section, do you have an ethic protocol reviewed by an ethic committee? 

Thanks 

Reviewer 2 Report

Comments and Suggestions for Authors

Intersting and well designed work despite the small number of subjects in the groups.

EDSS resulted  very low despite the disease duration, can the Authors give more information regarding clinical charactheristics of patients enrolled, in particular when previously treated with natalizumab  (symptoms at disease onset, EDSS during relapse)?

Data collection is from 10 patients under fingolimod treatment, and half of them resulted non responders.  I suppose this is a selection from the wider group of patients on DMTS. It could be useful to know the total number of Fingolimod patients in the full cohort of treated patients and procedure of selection.

Efficacy data are collected at year 2, but the transcriptomic changes are reported at 6 and 12 months. Please specify time at onset of MRI and/or clinical activity in non responders.

In Table 2 the chart regarding patients on NEDA 3 status  is unnecessary and redundant, please delete and amend the title of the table.

If known, specify if changes in S1P signaling pathway affect or is mediated by all the subtypes of receptors, because this is an implication to possibly  traslate these observation on other S1P modulators actually on the market.

Comments on the Quality of English Language

Very easy to understand, no further comments on this issue
